# Application of Electron Paramagnetic Resonance in an Electrochemical Energy Storage System

**Xiancheng Sang, Xixiang Xu, Zeyuan Bu, Shuhao Zhai, Yiming Sun, Mingyue Ruan * and Qiang Li ***

College of Physics, Weihai Innovation Research Institute, Qingdao University, Qingdao 266071, China
* Correspondence: ruanmingyue@qdu.edu.cn (M.R.); liqiang@qdu.edu.cn (Q.L.)

**Abstract:** The improvement of our living standards puts forward higher requirements for energy storage systems, especially rechargeable batteries. Unfortunately, phenomena such as capacity failure, etc. have been major difficulties in the field of energy storage. Therefore, we need some advanced means to explore the reaction process and mechanisms of the cell. Electron paramagnetic resonance (EPR) has the advantages of a high sensitivity to electrons, lack of damage to samples, quantitative analysis, etc., which can make for a more in-depth exploration of most paramagnetic electrode materials and metal electrode materials. After a brief description of the principle of EPR, this review briefly summarizes the application of EPR to the characterization of transition metal oxide cathode and lithium metal anode electrode materials in recent years, such as showing how to study electrode materials by using EPR in situ and *operando*.

**Keywords:** electron paramagnetic resonance (EPR); rechargeable battery; electrode material; lattice oxygen; metal anode

## 1. Introduction

The demand for energy in the world is rising incrementally. Additionally, the technology of human society continues to develop. Therefore, it is of great significance for human development to seek environmentally friendly energy, reduce the use of fossil energy and reduce the emission of greenhouse gases such as carbon dioxide. The exploitation and development of new energy provides a legitimate solution. At the same time, the development of energy storage systems is also a major challenge. In the past few decades, researchers have conducted a large number of studies on energy storage systems (especially on metal-ion batteries such as lithium-ion batteries [1], sodium-ion batteries [2], potassium-ion batteries [3,4], magnesium-ion batteries [5], calcium-ion batteries [6], and aluminum-ion batteries [7,8], etc.). Among them, the lighter atomic weight (6.94), sufficiently small ionic radius [9], and the most negative potential ($-3.045$ V) of $Li^+$ ions allow them to have better electrochemical performance compared with other metal-ion batteries [10,11]. Meanwhile, the abundance of sodium resources makes sodium-ion batteries a better option in the context of scarce lithium resources (the content of lithium in the earth's crust is only 0.0065%). Hence, in this article, we focus on the content related to lithium and sodium ion batteries.

The main components of metal-ion batteries are cathode, anode, separator, and electrolyte. During the charging and discharging process of cells, the anode and cathode materials always undergo complex structural changes, which affect the electrochemical performance of cells. Of particular importance, the cathode material is the main factor limiting the performance of cells, and the progress in the research of cathode material can directly promote the progress of the battery evolution. Therefore, the research and development of cathode materials has a milestone significance for the world energy revolution. For example, layered oxides have obvious advantages such as easy processing, simple structure, and large specific capacity. Hence, the technique has been widely studied in lithium and sodium ion batteries.

Various techniques have been used by researchers to study the reaction mechanisms and failure mechanisms of batteries, especially cathodes., such as X-ray diffraction (XRD) [12,13], scanning/transmission electron microscopy (SEM/TEM) [14–17], X-ray photoelectron spectroscopy (XPS) [18], synchrotron radiation [19], Raman spectroscopy [13], etc. During the charging and discharging process of the battery, the occurrence of the redox reactions accompanies the transfer of unpaired electrons in transition metals (TM) in electrode material [20]. This provides us an opportunity to dig out more useful information about the reaction mechanism by using the electron paramagnetic resonance (EPR) technique, which is extremely sensitive to unpaired electrons. Usually, the EPR technique can be used to qualitatively and quantitatively detect the unpaired electrons contained in TM, such as V, Mn, Fe, Co, Ni, and free radicals, and to explore the structural properties of its surrounding environment. It is of great significance to the research of layered transition metal oxides and polyanionic compound electrode materials, especially in the exploration of anion redox in layered oxides. Simultaneously, by using EPR spectroscopy, the changes in the electrolyte during the charging and discharging process and the effect of different electrolyte additives on cell performance can be demonstrated. Moreover, the in situ EPR technique is easier to implement than other in situ characterization techniques (in situ TEM, in situ XAS, etc.). Therefore, the real reaction mechanism of the cell can be reflected more intuitively and realistically by using EPR technique.

Based on the important role of the EPR technique in the area of battery research, in this review, we will mainly outline the application of EPR in the characterization of lithium and sodium-ion battery materials. Firstly, we briefly introduce the principle of EPR spectroscopy, including the internal and external interactions of electron paramagnetic resonance and the introduction of the EPR spectrometer. Secondly, we describe the research status of EPR on key battery issues and the key technologies of EPR. Finally, the research on the use of in situ EPR and imaging to investigating the anode behavior of lithium batteries was demonstrated.

## 2. Principles of Electron Paramagnetic Resonance Technology

EPR is a technique belonging to spectroscopy. When the free electron is in the external magnetic field, the electron's energy level is split from the degenerate state into two energy levels. If we then add fixed frequency microwave radiation in the direction perpendicular to the external magnetic field, the free electron will absorb these microwaves under a specific external magnetic field strength, resulting in the phenomenon of transition from a low energy level to a high energy level:

$$\Delta E = h\nu = g_e \beta_e B_0 \tag{1}$$

where $h$ is the Planck's constant, $\nu$ is the microwave frequency, $g_e$ is the electron's Lander factor, $\beta_e$ is the electron's Bohr magneton, and $B_0$ is the external magnetic field strength.

The research objects of EPR mainly include the following three categories: (1) free radicals, including monoradicals, biradicals, polyradicals, triplet molecules and gas molecules ($O_2$, $CO_2$, $NO_2$) [21]; (2) transition metal ions and rare earth metal ions containing unpaired electrons in atomic orbitals, such as $Co^{2+}$ ($3d^7$), $Fe^{3+}$ ($3d^5$), $Fe^{2+}$ ($3d^6$), $Mn^{2+}$ ($3d^5$), $Cu^{2+}$ ($3d^9$), $Cu^+$ ($3d^{10}$) and $V^{4+}$ ($3d^1$) [22,23]; (3) defective bits, such as either paramagnetic free radicals, oxygen defects, and metal ions containing unpaired electrons which are generated during the electrochemical reaction of most electrode materials. It is worth noting that the gyromagnetic ratio of free electrons is relatively large, so EPR has the advantage of high sensitivity that nuclear magnetic resonance (NMR), which is also a magnetic resonance technology, does not have.

Interactions in EPR can be divided into internal interactions and external interactions. The former consists of the interaction between electron spin and the magnetic field generated by the sample itself; the common ones include electron-nucleus hyperfine interaction (HFC) and interactions between electrons ($D_e$). The latter is the interaction of electron spin with the magnetic field generated by the external instrument, mainly including the

electron Zeeman interaction ($Z_e$). The Zeeman interaction ($Z$) and the electric quadrupole moment interaction ($Q$) of the nucleus also appear in EPR, although these two interactions are usually so small, we can ignore them. The Hamiltonian of EPR can be written as:

$$H_e = H_{HFC} + H_{D_e} + H_{Z_e} + H_Z + H_Q \qquad (2)$$

If a single electron is surrounded by a nucleus with spin $I$, the electron-nucleus hyperfine interaction is:

$$H_{HFC} = S \cdot \overline{\overline{A}} \cdot I \qquad (3)$$

$S$ is the electron spin quantum number and $\overline{\overline{A}}$ is a tensor. It is apparent that the magnetic moment is affected by the magnetic field created by a nearby magnetic moment. As a result, hyperfine interactions arise.

Zero-field splitting is well known as an interaction, one which refers to the energy level splitting in the absence of an external magnetic field. The cancellation of electron orbital degeneracy is known as the direct cause of zero-field splitting. In the zero field state in some multi-electron systems (such as TM elements), the trend of electron arrangement $S > \frac{1}{2}$ appears due to the strong interaction between electrons, leading to the cancellation of electron orbital degeneracy. At this time, the orbital energy level is redistributed and the energy level splitting occurs:

$$H_{D_e} = S \cdot \overline{\overline{D}}_e \cdot S \qquad (4)$$

$\overline{\overline{D}}_e$ is called the fine structure tensor. The existence of multiple unpaired electrons in the $d$ and $f$ orbitals of TM leads to giant ZFS effect. In addition, for the case where $S$ is an integer, the parallel mode EPR has a high probability of detecting forbidden transitions.

The electron's Zeeman interaction can be expressed as:

$$H_{Z_e} = \frac{\beta_e}{h \overrightarrow{B}_0} \cdot \overline{\overline{g}} \cdot \overrightarrow{S} \qquad (5)$$

where $S$ means electron spin quantum number and $\overline{\overline{g}}$ is a tensor. However, in most cases, the additional magnetic and electric fields generated by multiple unpaired $d$ electrons in transition metal ions (such as high-spin $Mn^{2+}$ or $Fe^{3+}$) lead to further fine splitting of energy levels, which makes the EPR spectrum more complicated.

For transition metal ions such as $Mn^{2+}$, the electron Zeeman effect is smaller than the interaction between other electrons. At the critical moment, the energy of the microwaves may not be sufficient to excite all transitions and only the lowest spin states will be thermally populated. Only the ground-state EPR spectrum of the paramagnetic system can be observed if the energy level splitting is much larger than $k_B T$ at this time.

In the specific structure of the TM-ion, the arrangement of unpaired electrons in the electron's orbitals is determined by the relative magnitudes of the electron orbital splitting energy $\Delta$ and the electron pairing energy $p$. In a weak crystal field, $\Delta < p$; at the time, the electrons are more inclined to arrange within different orbitals, as shown in Figure 1a,b, where the ion is in a high spin state. In a strong crystal field, $\Delta \gg p$, electrons preferentially spin opposite pairing in $t_{2g}$ orbitals; the ion is in a low-spin state, as shown in Figure 1c,d. Therefore, the same ion may exhibit different EPR states in different crystal fields [24,25]. Taking $Fe^{2+}$ of $3d^6$ as an example: in a strong crystal field, it is in the low-spin state of $S = 0$, which is the EPR-silent state; in a weak crystal field, it is in the high-spin state of $S = 2$, which is the EPR-active state. Whereas the $Fe^{3+}$ of $3d^5$ is always in the EPR-active state, whether it is at low-spin state with $S = 1/2$ or high-spin state with $S = 5/2$.

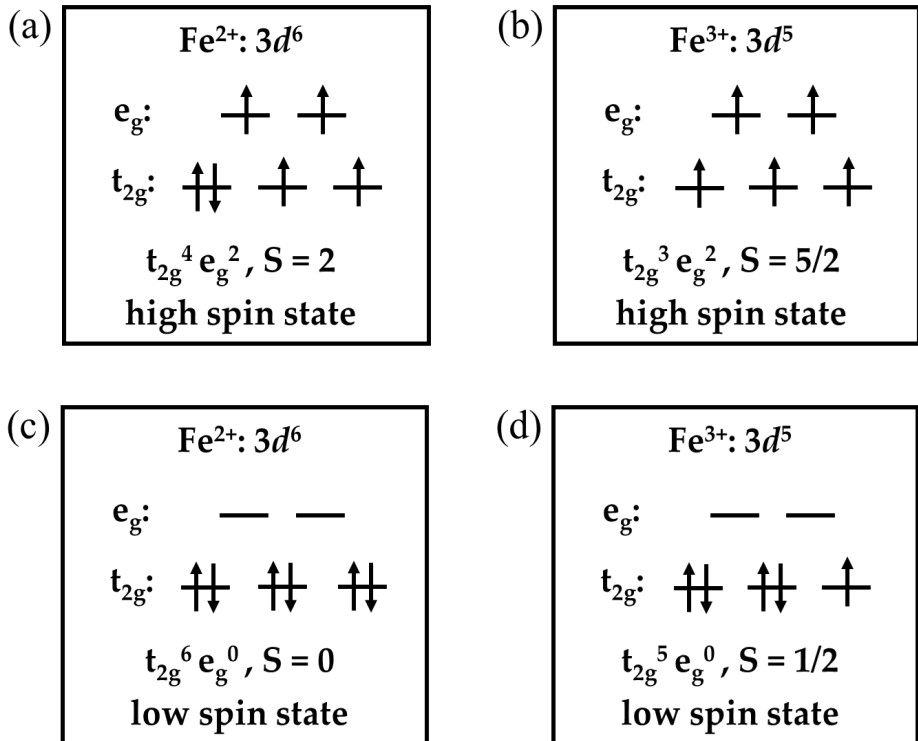

**Figure 1.** The arrangement of $3d$ electrons in high-spin state (**a**,**b**) and low-spin state (**c**,**d**) of $Fe^{2+}$ and $Fe^{3+}$.

A common EPR spectrometer consists of a microwave bridge, microwave source, magnetic field, resonator, and other components. The commonly used microwave bands are: X-band (~9.5 GHz), K-band (~24 GHz), Q-band (~35 GHz), and W-band (~94 GHz). In general, the waveguide and resonator sizes of the most widely used X-band instruments are suitable for polycrystalline, solution, or powder samples. The resonator of the Q-band instrument is smaller, and the filling coefficient of the sample in the cavity is larger. Thus, it is suitable for small single-crystal samples. It is noteworthy, one of the advantages of the Q-band instrument is that the required amount of cold liquid (such as liquid helium) is relatively rare during the low-temperature measurement.

### 3. Application of EPR Spectroscopy in a Rechargeable Battery System

*3.1. Exploring the Redox Mechanism in Transition Metal Oxide Cathode Materials*

The improvement of cathode materials' performance is pivotal to improving the overall performance of the metal-ion batteries. In Li/Na-ion batteries, layered oxide cathode materials can generate additional capacity due to their reversible anionic (such as lattice oxygen) redox, resulting in a higher specific capacity. Therefore, the research on the anionic redox mechanism has received extensive attention [26–31]. However, there are few methods to characterize the redox reaction of lattice oxygen at present, and EPR can accurately monitor the change of oxygen species between molecular oxygen and $O^{2-}$ during the cell charging and discharging process, which provides great opportunities for studying the oxygen redox mechanism in electrode materials.

The g-factor of 2.006 for oxidized oxygen species ($O_2^{n-}$, n = 1, 2, 3) in layered oxide cathodes for Li-ion and Na-ion batteries is well-recognized [32–35]. Liu et al., have performed extensive research, revealing the oxidized oxygen state in Li $3d$ oxide cathodes with oxygen redox activity (including O3-structured $Li_{1.2}Ni_{0.2}Mn_{0.6}O_2$ and $Li_{1.2}Ni_{0.13}Co_{0.13}Mn_{0.54}O_2$, O2-structured $Li_{1.033}Ni_{0.2}Mn_{0.6}O_2$, and disordered rock-salt ($Li_{1.2}Ti_{0.4}Mn_{0.4}O_2$) by non-invasive EPR techniques [36]. Molecular $O_2$ signals can be seen in the EPR curve of $Li_xNi_{0.2}Mn_{0.6}O_2$ and $Li_xNi_{0.13}Co_{0.13}Mn_{0.54}O_2$ when the cell is fully charged, and it has been demonstrated that the molecular $O_2$ gradually decreases in the materials during discharge; it then appears reversibly in subsequent cycles. Surprisingly, the vacancy left after TM

migration is the place where this molecular oxygen is bound. This is where it differs from gaseous oxygen [37–40]. Notably, after the first cycle, the order of Li and TM in the material was rearranged, and the birth of molecular oxygen became easier.

In sodium-ion batteries, Zhao et al., used EPR to reveal for the first time the coexistence of $O_2^{n-}$ (n = 1, 2, 3) and trapped molecular $O_2$ during the oxygen redox process of P2-structured $Na_{0.66}Li_{0.22}Mn_{0.78}O_2$ [41]. Meanwhile, in the process, molecular $O_2$ was determined to be trapped in the "desodiumated" electrode material.

Unfortunately, the reversibility of oxygen redox in layered oxide materials is usually limited, and it cannot return to its natal state as expected. There is always some molecular $O_2$ diffusing out of the material and causing irreversible capacity loss. Therefore, researchers have used many methods to suppress the overflow of molecular $O_2$, such as trace anion substitution [40], transition metal substitution [42], inhibition of oxygen redox activity [43], doping, surface repair co-modification [44], etc.; it can clearly be seen that the reversibility of oxygen redox is greatly improved by EPR.

In terms of anion substitution, the authors improved the reversibility of oxygen redox by substituting a very small amount of O with F, as shown in Figure 2.

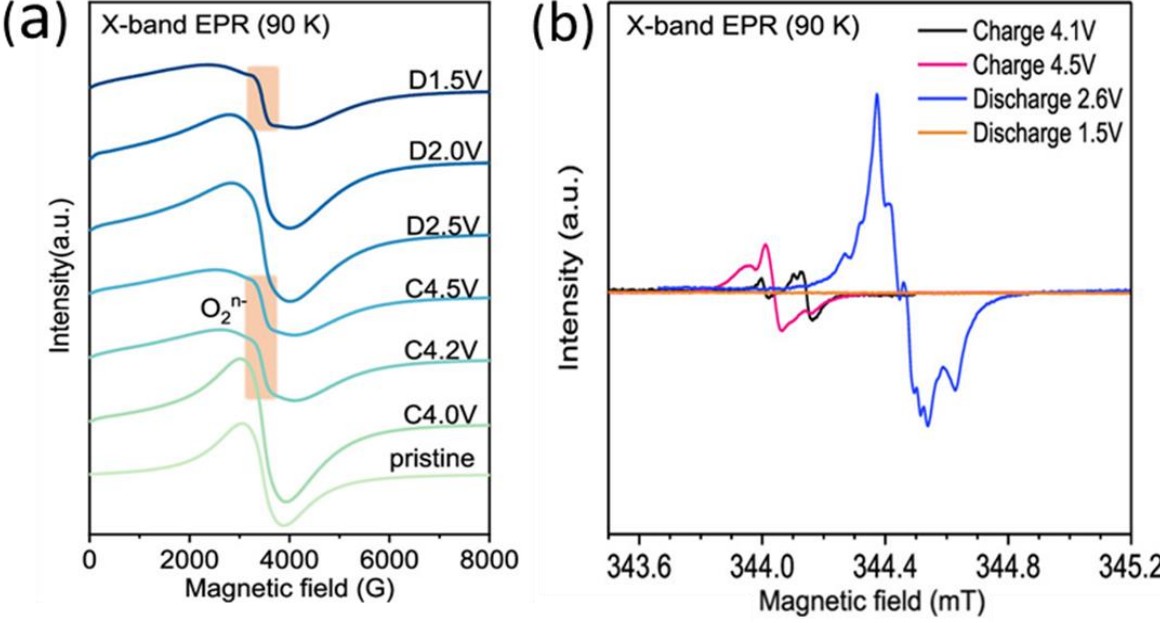

**Figure 2.** (**a**) EPR measurements of F-substituted $Na_{0.65}Li_{0.22}Mn_{0.78}O_{1.99}F_{0.01}$ at different charge-discharge states in the first cycle at 90 K. The pristine NLMOF exhibits only spectral lines representing antiferromagnetically coupled of $Mn^{4+}$ ($t_{2g}^3 e_g^0$, $S$ = 3/2). When charged to 4.2 V, a signal of oxidized oxygen species $O_2^{n-}$ (n = 1, 2, 3) appears, indicating that the oxygen in the material is oxidized. Meanwhile the line-shape of the $Mn^{4+}$ EPR signal becomes wider. At the end of charging, the line-shape of $Mn^{4+}$ is further broadened. The appearance of this phenomenon may be caused by the stronger electron-electron dipole interaction between $Mn^{4+}$ and $O_2^{n-}$. During the discharge process, the spectral line of $Mn^{4+}$ returned to its original state, and the signal of $O_2^{n-}$ seemed to disappear between 2.5 and 2.0 V. At this time, $O_2^{n-}$ was successfully reduced. Additionally, because of the strong electron-electron dipole interaction of $Mn^{3+}$, the EPR spectral line is wider from 2.0 to 1.5 V. It is worth noting that the $O_2^{n-}$ signal at 1.5 V originates from carbonate species generated during discharge. Panel (**a**) is reproduced with permission from ref. [40]. Copyright 2021, copyright American Chemical Society. (**b**) Similar EPR measurements were performed on unsubstituted $Na_{0.72}Li_{0.24}Mn_{0.76}O_2$ in the region where the oxygen resonance peak appears. The hyperfine structure of the EPR signal of $O_2^{n-}$ is shown here. Panel (**b**) is reproduced with permission from ref. [35]. Copyright 2020, copyright American Chemical Society.

It can be seen from Figure 2 that the EPR signal of $O_2^{n-}$ in NLMO exhibits various hyperfine structures, but not in NLMOF. This suggests that substituting a trace amount of O with F can greatly alleviate the anisotropic coupling between $Mn^{4+}$ and $O_2^{n-}$, which in turn promotes the reversible oxygen redox reaction. Wang et al., have prepared $Li_{1.1856}K_{0.0119}Ni_{0.2000}Mn_{0.5960}O_2$ (KLNMO) by $K^+$ doping $Li_{1.2}Ni_{0.2}Mn_{0.6}O_2$ (LNMO) [45]. Subsequently, as the cycling process progresses, the $Mn^{4+}$ EPR signal intensity of KLNMO decays relatively slowly compared with LNMO. This result proves that $K^+$ doping effectively alleviates the migration of TM ions and oxygen vacancies, resulting in improved structural reversibility.

As mentioned earlier, molecular $O_2$ is generated during charging and is usually trapped in the holes left after in-plane TM migration. Here, the migration of TM is involved and will inevitably lead to different degrees of metal clusters, thereby affecting the electrochemical performance of cells. Therefore, exploring the distribution and reaction process of transition metals in electrode materials not only guides the design of high-performance materials but also enables people to gain a clearer understanding of the reaction mechanisms of electrode materials. Related research was carried out by Labrini et al., using EPR [46].

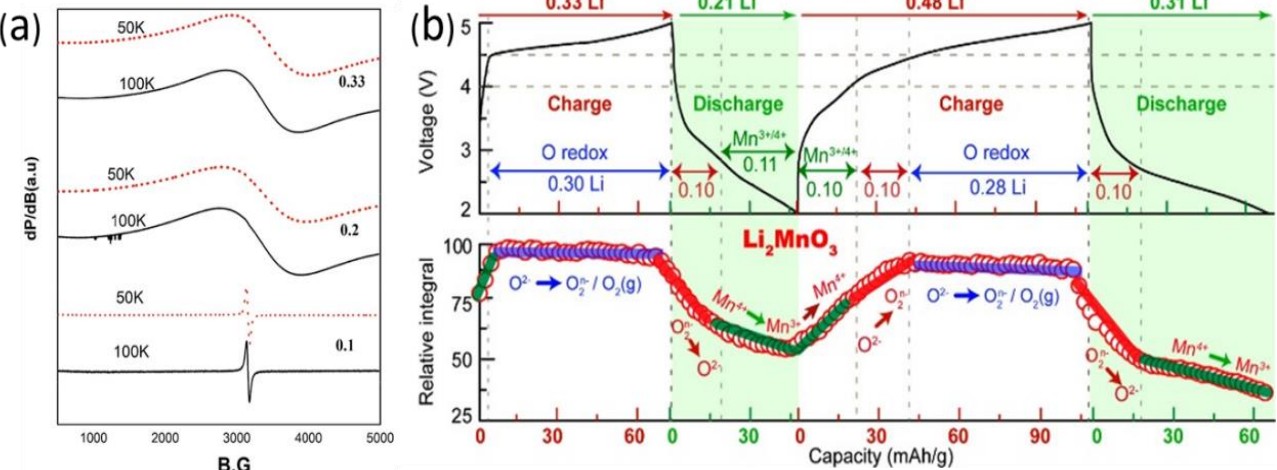

**Figure 3.** (**a**). EPR test of $LiNi_xCo_{1-2x}Mn_xO_2$ at 50K and 100K. From top to bottom x is equal to 0.33, 0.2, 0.1, respectively. (The $Mn^{4+}(t_{2g}{}^3e_g{}^0)$ signal is shown here in X band.) Panel (**a**) is reproduced with permission from ref. [46]. Copyright 2019, copyright Elsevier. (**b**) Evolution of the $Li_2MnO_3$ cathode EPR signal integral with charge-discharge voltage in a $Li_2MnO_3$/Li half-cell. $Mn^{4+} \leftrightarrow Mn^{3+}$ (green), O redox at high potential (blue), and reversible $O^{2-} \leftrightarrow O_2^{n-}$ (red) at high potential (measured at X band). Panel (**b**) is reproduced with permission from ref. [47]. Copyright 2017, copyright American Chemical Society.

It can be seen from Figure 3a that the EPR spectral broadening of the $LiNi_{0.1}Co_{0.8}Mn_{0.1}O_2$ phase is significantly narrower than that of the other two phases. This may be because there are fewer $Mn^{4+}$-$Ni^{2+}$ clusters in the sample, which makes the dipole-dipole interaction weaker. While in the other two phases, different $Mn^{4+}$ ions merge into a broad EPR signal with linewidth proportional to the average Ni-Mn ratio in the first coordination layer of $Mn^{4+}$. Compared with the $LiNi_{0.2}Co_{0.6}Mn_{0.2}O_2$ phase, further increasing the Ni content in the material makes $LiNi_{0.33}Co_{0.33}Mn_{0.33}O_2$ produce some ($Ni^{2+}$(slab)-$Ni^{2+}$(interslab)) clusters in the long-distance range, which results in a stronger magnetic exchange interaction. The linewidth of the EPR spectral line is reduced to a certain extent. Labrini et al., also explained the increase of EPR signal line width during the charging process through $Li_yCo_{0.8}Ni_{0.1}Mn_{0.1}O_2$ not only $Ni^{2+}$ is oxidized to $Ni^{3+}$, but $Co^{3+}$ is also oxidized to $Co^{4+}$ during the delithiation process [48].

Before that, Tang et al., explored the quantitative changes of anionic and cationic redox reactions in cathode materials such as $Li_2MnO_3$ through operando EPR [47]. The authors calculated the EPR signal integrated of the $Mn^{4+}$ to reflect the redox state and degree of each component in the material during the electrochemical reaction, as shown in Figure 3b. The

increase in the integral intensity of the EPR signal during charging is due to the oxidation of a small amount of $Mn^{3+}$ remaining in the original material to $Mn^{4+}$. During further charging, the signal of $Mn^{4+}$ remains basically unchanged. This proves that there is no redox process for the Mn element in this process. The authors attribute this process to the oxidation reaction of anions. During the first discharge process, $O_2^{n-}$ is first reduced to $O^{2-}$, and $Mn^{4+}$ is directly reduced to $Mn^{3+}$ below 3.0 V. It is worth noting that the EPR signal intensity is lower after the second discharge than after the first discharge, which indicates that more $Mn^{4+}$ is involved in the subsequent reduction reaction. This corresponds to the increase in capacity during the second charge-discharge cycle. Apart from this, the authors performed the same examination on $Li_{1.2}Ni_{0.2}Mn_{0.6}O_2$ and $Li_{1.2}Ni_{0.13}Mn_{0.54}Co_{0.13}O_2$. No further description is given here.

In addition, the dissolution of transition metal ions in the cathode material is also one of the causes of degradation of the performance of cells [49,50]; the dissolved transition metal ions may also lead to electrolyte decomposition. Szczuka et al., used pulsed EPR to reveal that the complexation of vanadium ions formed after the dissolution of $V_2O_5$ would complex with the carbonate products degraded in the electrolyte. Huang et al., compared the degree and difficulty of dissolution of metal ions in LMO by using electrolytes containing different anions [50]. The electrolyte containing $PF_6^-$ and $ClO_4^-$ significantly accelerated the dissolution of Mn ions compared to the electrolyte containing only $TFSI^-$.

### 3.2. Explore the Reaction Mechanism of Polyanionic Compounds Cathode Materials

Polyanionic compounds have attracted extensive attention from researchers due to their excellent cycling performance and rate capability brought about by their stable three-dimensional structure [51–54].

Similarly, we can also explore the reaction mechanism of polyanion-based cathode materials by EPR spectroscopy. For example, Li et al., revealed the charge compensation mechanism of $Na_3V_2(PO_4)_2O_{1.6}F_{1.4}$ during charging [55]. It is different from the previously reported low-valent vanadium ions that are completely oxidized to $V^{5+}$ [56] in that only $V^{4+}$ is oxidized to $V^{5+}$ [57]. Its reaction mechanism is that part of $V^{3+}$ is oxidized to $V^{4+}$, and most of $V^{4+}$ is further oxidized to $V^{5+}$, and the material is in a mixed solid solution phase containing $V^{3+}/V^{4+}/V^{5+}$ when fully charged. As we can see from Figure 4: The line-widths of the EPR spectra of $Na_3V_2(PO_4)_2O_{1.6}F_{1.4}$ are always at a similar level when the cell is cycling, on account of the fact that the $V^{3+}$ center is always present during Na ion insertion/extraction.

However, since the unique polyanionic structural units in polyanionic cathode materials are tightly connected by strong covalent bonds, the valence electrons of polyanionic groups and transition metal ions are isolated from each other. This reduces the electronic conductivity of the material. In turn, the electrochemical properties such as the rate capability of the material are inevitably reduced. For this reason, various strategies such as nanosizing, carbon encapsulation, and foreign ion doping have been attempted to modify polyanionic compounds. $F^-$ substitution is a good way to improve the electrochemical properties of polyanion compounds. Rubio et al., found that, given the change of EPR signal from the $VO^{2+}$ defect due to the substitution of $O^{2-}$ for F- in the charging and discharging process of the magnesium ion battery, it is proved that the multi-electron electrochemical reaction from $Na_5V(PO_4)_2F_2$ $V^{3+}$ to $V^{5+}$ during the charging process and the reaction process of $Mg^{2+}$ ions preferentially coordinated with $O^{2-}$ during the discharge process by EPR [58].

The signal of $VO^{2+}$ without hyperfine structure appears in the EPR spectra of $Na_5V(PO_4)_2F_2$ at different charge states, as shown in Figure 5a. This indicates that vanadyl ions are strongly coupled by exchange interactions. The signal intensity of $VO^{2+}$ decreased when more than one $Na^+$ was extracted, which means that the diamagnetic $V^{5+}$ may appear at this time. The multi-electron electrochemical reaction of $V^{3+} \leftrightarrow V^{5+}$ during the charge-discharge process of $Na_5V(PO_4)_2F_2$ was demonstrated. Interestingly, the signal of $VO^{2+}$ does not return to the initial

level after full discharge. The stability of VO$^{2+}$ is destroyed due to the preferential coordination with O$^{2-}$ in the process of Mg$^{2+}$ insertion.

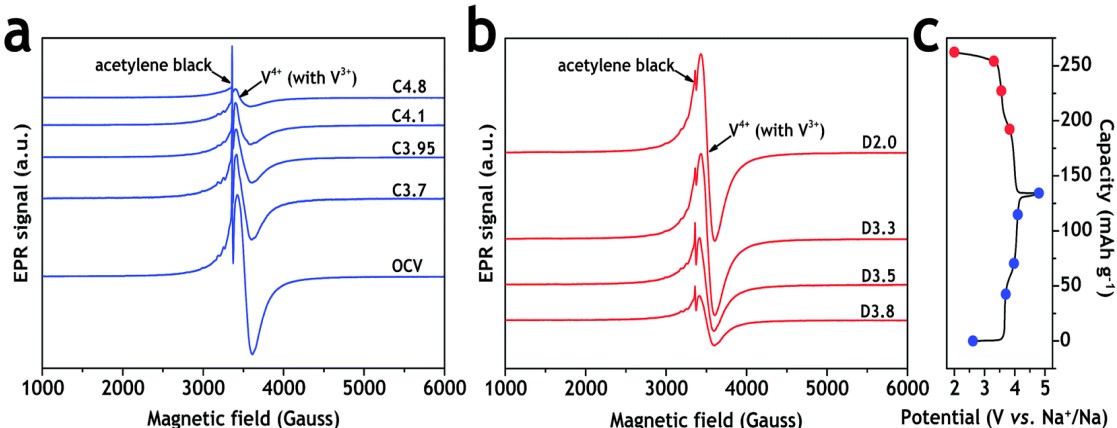

**Figure 4.** (**a**) EPR spectrum of Na$_3$V$_2$(PO$_4$)$_2$O$_{1.6}$F$_{1.4}$ during the charging process. The EPR signal representing V$^{4+}$ (t$_{2g}^1$e$_g^0$, $S$ = 1/2) gradually diminishes during charging and does not disappear completely when charging to 4.8 V. (**b**) EPR spectrum of Na$_3$V$_2$(PO$_4$)$_2$O$_{1.6}$F$_{1.4}$ during the discharging process. The EPR signal of V$^{4+}$ is gradually enhanced back to the initial level during the discharge. The spin state of V$^{3+}$ in the spectrum is t$_{2g}^2$e$_g^0$, $S$ = 1. Measured at X band. (**c**) The blue and red dots correspond to the potential of the battery when the spectral line in figure (**a**,**b**) is obtained. Reproduced with permission from ref. [55]. Copyright 2018, copyright Royal Society of Chemistry.

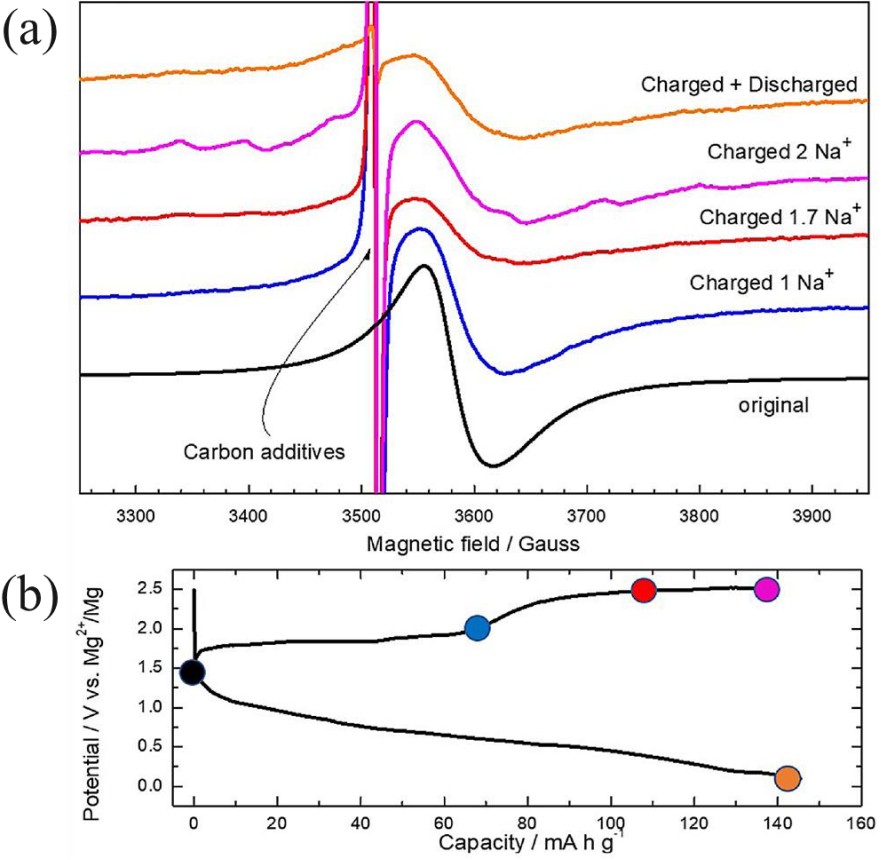

**Figure 5.** (**a**) The original EPR spectra of Na$_5$V(PO$_4$)$_2$F$_2$ and the corresponding EPR spectra under different sodium extraction amounts and complete discharge (measured at X band); (**b**) corresponding potential under different sodium extraction amount. Reproduced with permission from ref. [58]. Copyright 2021, copyright Elsevier.

### 3.3. Study of the Anode State in Electrochemical Cycle

Graphite is extensively used as an anode material for lithium-ion batteries; its strong electrical conductivity and theoretical specific capacity of 372 mAh g$^{-1}$. During electrochemical cycling, lithium is more than intercalated between the graphite layers, it is also partially deposited on the graphite to form the solid electrolyte interface (SEI). Inevitably, dead lithium will be generated, which will affect the electrochemical performance and accelerate the degradation of the anode. We therefore need realistic, non-destructive means to quantify lithium intercalation and deposition in graphite. In this way, we can prevent the degradation of the anode through some scientific means. Wang et al., characterized the lithiation and lithium deposition of graphite anodes during the cycling of three-electrode batteries by using in situ EPR [59]. The authors demonstrated the formation of SEI films at around 1.3 V by the variation of the conductivity of lithiated graphite. Since lithiated graphite and deposited lithium possess different EPR line-shapes (2.5 G and 1.2 G line-widths, respectively), it is possible to accurately distinguish the deposition potential of lithium and quantify the generation of dead lithium on the graphite anode. Additionally, the VC additive can effectively inhibit the formation of metal Li0 on the graphite anode by comparison.

Due to the current demand for high energy density batteries in various emerging fields. Anode-free Li-metal batteries (AF-LMB) have attracted widespread attention for the significant increase in energy density attained by their elimination of the use of virgin anode materials, while significantly reducing the production cost of the batteries [60–64]. Anode-free Li-metal batteries (AF-LMB) have significantly improved the energy density and significantly reduced the production cost of batteries by not using the original anode material, which has attracted widespread note [60–64]. This is an ideal high-energy density energy storage system There is no doubt that the performance of AF-LMB is significantly affected by the deposition state of lithium on the anode substrate. Therefore, exploring changes in the deposition state of metallic lithium is of great significance to the development of high-energy density batteries.

At X-band frequencies, the skin effect limits the penetration of microwaves to only about 1.1 μm of the Li metal surface, so different lithium forms have different EPR line-shapes [65], as shown in Figure 6:

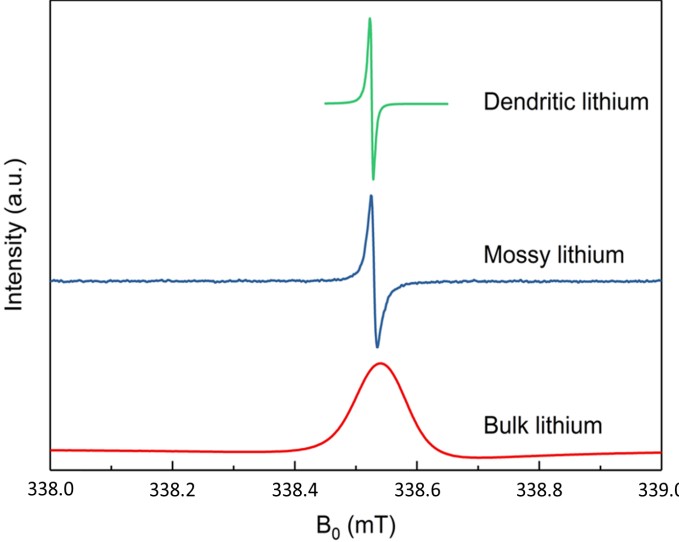

**Figure 6.** First derivative CEPR signal, as measured in a field-swept EPR experiment examining metallic lithium with different morphologies. The peak-to-peak line-width is minimal for dendritic lithium with ca. 0.005 mT (green), showing a Lorentzian line-shape. For mossy lithium, it increases to 0.03 mT (blue). It reaches a maximum for bulk lithium with ca. 0.15 mT (red), showing a Dysonian line-shape. Reproduced with permission from ref. [65]. Copyright 2018, copyright Springer Nature.

Neimöller et al., explain how different EPR line-shapes can be used to distinguish between different species of lithium forms. The EPR spectrum of dendritic lithium shows the most symmetric and narrow Lorentzian line-shape. This indicates that the size of dendritic lithium is much smaller than the skin depth and that only one metallic lithium species contributes to the signal, that is, all lithium is dendritic. Mossy lithium shows a Lorentzian line-shape with a slight phase shift due to the fact that the effective thickness of mossy lithium is slightly higher than the skin depth of microwaves. In sum, whether it is dendritic or mossy lithium, they all have the same scale as skin depth. In contrast, the space for free movement of conducting electrons in bulk lithium metal is greatly relative. Then, as the block increases, the ratio of the diffusion time of electrons in the skin depth of metal surface $T_D$ to spin-spin interaction relaxation time $T_2$ decreases, which causes the Lorentzian line-shape to gradually become a Dysonian line-shape.

Likewise, monitoring the deposition of the anode metal is of paramount importance for breakthroughs in ion battery technology; electron paramagnetic resonance imaging (EPRI) technology is also a very intuitive means to explore microscopic lithium morphology [66]. Geng et al., demonstrated the deposition of lithium on the anode copper foil in the first two cycles by studying the LCO||Cu cell [67].

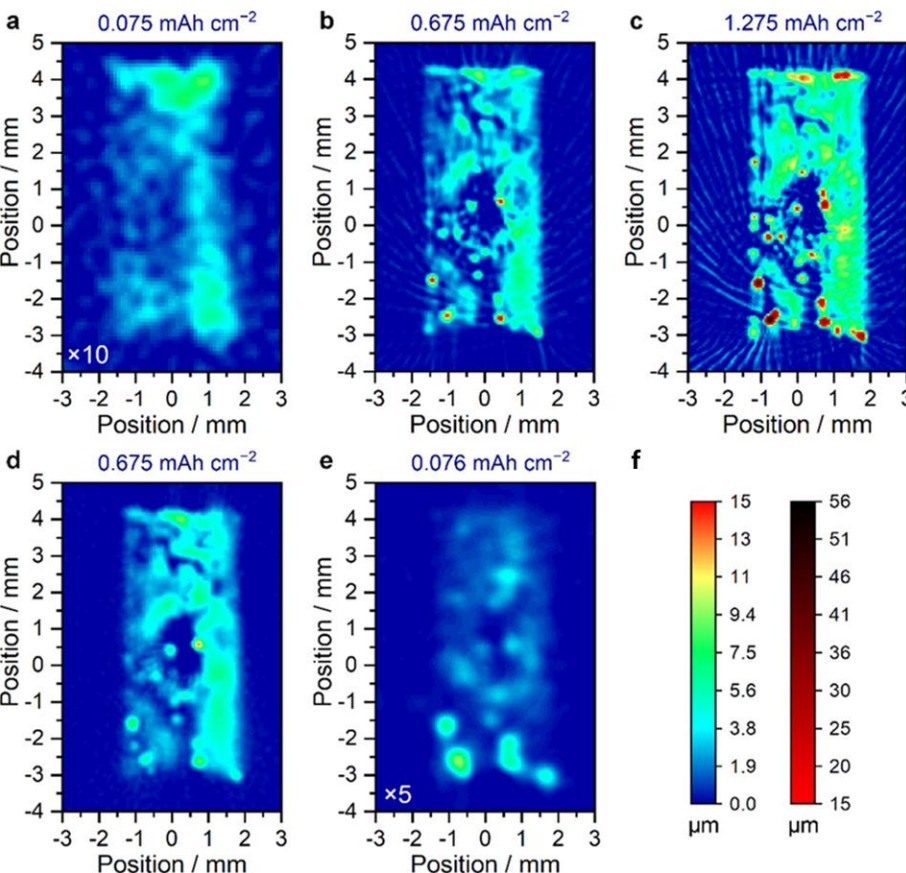

**Figure 7.** In situ EPRI for the first cycle of LCO||Cu cell. (**a**–**c**) Images at the end of three charging phases (1Ci, 1Cii, and 1Ciii); (**d**,**e**) images at the end of two discharging phases (1Di and 1Dii) (measured at X band); and (**f**) color bars representing the thickness values of Li deposition. Reproduced with permission from ref. [67]. Copyright 2021, copyright American Chemical Society.

As shown in Figure 7: In the initial stage of charging to 0.075 mAh cm$^{-2}$, metallic lithium preferentially nucleates in the edge region of the copper foil while the central part is blank, and the deposited thickness of Li in the upper right part is ≈0.87 μm. Further charging to 0.675 mAh cm$^{-2}$, the appearance of dendritic/mossy lithium leads to local excessive deposits (LEDs) at the red sites. It is demonstrated that lithium is preferentially

deposited at nucleated sites. Charged to 1.275 mAh cm$^{-2}$, lithium plating becomes more uniform, as the amount of lithium increases. When discharged to a 1Di state with the same amount of lithium deposited as 1Cii, more uniform Li deposition than 1Cii can be seen as well as preferential Li stripping on the vast majority of the LEDs. In further discharge to the 1Dii state, contrasting with 1Ci, the remaining dead Li is more likely to be found where the LED is located. This represents the lithium microstructure at the LED, which will unavoidably undergo fracture during the lithium stripping process. Additionally, the authors found in the second cycle that lithium was deposited more preferentially at the location of dead lithium rather than at the location where lithium had been uniformly deposited and stripped. This provides some new ideas for the development of the anode-free cell, for instance, regulating the size and the sharpness of the edge of the copper foil, and controlling the lithium metal deposition to ensure stronger performance.

## 4. Conclusions

This paper briefly expounds on the basic principles of EPR and the application of EPR spectroscopy in energy storage systems in recent years. As we all know, to promote the development of batteries in the direction of high energy density, the pursuit of high-performance cathode materials has never ended. Layered transition metal oxide materials have received much attention in lithium and sodium ion batteries. EPR spectroscopy has unique advantages for the research of layered transition metal oxides due to its high sensitivity to TM and oxidized oxygen species $O_2^{n-}$. Additionally, the technique makes it easier to obtain the material reaction mechanisms difficult to obtain by general characterization methods, such as the respective reaction sites and capacity contributions of transition metal ions and $O_2^{n-}$. The formation of gaseous oxygen and the dissolution of transition metal ions during the electrochemical reaction can cause irreparable loss of battery performance. Demonstrating these negative behaviors through the use of EPR spectroscopy will provide us with an array of ideas for designing better materials. For example, anion/cation substitution, surface modification, etc., can also guide us in selecting more suitable additives for an electrolyte. However, the unparalleled sensitivity of EPR to electronic structure requires us to use some other characterization tools to assist it. Because of the defects in the material and the local non-uniform composition distribution, the EPR measurement results will be greatly affected. For example, we can use XANES to determine the average oxidation state of TM. The relative distribution of oxidation states was determined by EPR. In addition, we briefly described the research on the reaction mechanism of EPR on polyanionic compounds (mainly $Na_3V_2(PO_4)_3$ series substances). EPR revealed the state of TM ions in polyanionic compounds during charge and discharge. It has been proved that ion doping and fluoride ion substitution are feasible means for the improvement of the electrochemical performance of polyanionic compounds. On the other hand, in-situ EPRI characterization and quantification of anode lithium deposition provide us with a new perspective. Moreover, the influence of different electrolyte additives on the formation of dead lithium can be revealed in real time by the use of EPR. In the future, some guiding suggestions for material modification and electrolyte selection, etc. can be provided through EPR spectroscopy. It should be noted that the materials used for cell construction should not reflect microwave radiation and should be EPR-silent state. Alternatively, we could clearly distinguish the "impurities" in the EPR measurement, such as the sharp carbon signal, so as not to interfere with our measurement.

**Author Contributions:** Conceptualization, Q.L.; investigation, X.S. and X.X.; writing—original draft preparation, X.S., S.Z. and Y.S.; writing—review and editing, M.R., X.S. and Z.B. All authors have read and agreed to the published version of the manuscript.

**Funding:** This work was supported by the National Natural Science Foundation of China (22179066), and the National Science Foundation of Shandong Province (ZR2020MA073).

**Institutional Review Board Statement:** Not applicable.

**Informed Consent Statement:** Not applicable.

**Data Availability Statement:** The data presented in this study arises from the sources cited or are available on request from the corresponding authors.

**Acknowledgments:** The authors would like to thank Mingyue Ruan and Qiang Li for extremely helpful discussions.

**Conflicts of Interest:** The authors declare no conflict of interest. The funders had no role in the design of the study; in the collection, analyses, or interpretation of data; in the writing of the manuscript; or in the decision to publish the results.

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
