# Peer review of "Application of Electron Paramagnetic Resonance in an Electrochemical Energy Storage System"

_magnetochemistry, doi:10.3390/magnetochemistry9030063_

Round 1

Reviewer 1 Report

Li and co-authors have prepared a short review on the application of EPR in studying batteries with a focus on the paramagnetic centers founds on electrode materials. The use of EPR to study energy systems has received much attention in recent years making this a topical contribution and appropriate for the journal. Nevertheless, some revisions are needed before it can be accepted. 

In the introduction (line 51-53). general references are needed for the other techniques that have been used to study batteries.

The description of EPR theory is rather poor (line 96-126). For example the authors confused zero-field interactions with electron-electron dipolar interactions. An EPR text book or handbook such as "EPR spectroscopy: fundamentals and methods" edited by Goldfarb and Stoll needs to be consulted. This section needs heavy editing and references which give much more in depth information added to aid the curious reader.

A better introduction of low spin and high spin TM systems (lines 88-89, 127-134) and their nomenclature (eg t2g3eg0) should be given because otherwise the manuscript becomes disjointed for an none expert reader. Spin states S=1/2, S=3/2 etc should be explicitly given.

A brief description of the state of the art of EPR spectrometers, including the microwave frequencies now available and in situ/in operando equipment would be helpful. Include recent review papers where these are discussed in more detail will also be helpful for the curious reader. A description of multifrequency EPR will also help the authors to also clear up their description of high spin systems and what can be measured by EPR at a given frequency/field.(line 122-126). 

Related to the above, the microwave frequency (X-band, Q-band etc) should be given in all figure captions. It is missing in many cases.

In the discussion of the literature examples, it would be useful to include the spin-state for the observed EPR active species. 

"And it is proposed to use auxiliary means to improve the accuracy of EPR signals.." line 18, where is this discussion in the main text other than a single sentence about comparing results with XANES in the conclusion?

Overall the the grammar was passible, but some very odd or incorrect word have been used. For example line 161: "...the de-sodomized electrode material." or line 236 "...the sinners that destroy the good performance of cells."

Furthermore, the authors appears to have been rather careless with the editing, particular with respect to scientific writing conventions such as line 143 "molecular oxygen and O2- during", line 241 "PF6— and ClO4—" and throughout there are no space between quantity and units.

A native English speaking chemist needs to carefully proofread this manuscript.

Reviewer 2 Report

Summary:

The focus of this review is on the use of in situ and operando EPR spectroscopy and imaging for the characterization of transition metal oxide cathode and lithium metal anode materials in rechargeable battery cells. The authors also suggest ways to increase the EPR technique's accuracy to enhance electrode material design. Authors mainly focus on using EPR in the characterization of lithium and sodium-ion battery materials.

Minor errors:

Line 18: Herein, the sentence cannot start with „And“.

Line 55: reaction -> reactions

Line 56: It -> This

Lines 72 and 73: Should be one sentence.

Lines 90-92: Should be rephrased.

Line 96: Needs plural.

155-156: Should be rephrased.

Line 176: As same as in Line 18.

Line 222: integrated strength of the EPR signal -> EPR signal integral

Line 230: weaker -> lower

Line 267, 378, 382, 385: As same as in Line 18.

General impression:

This manuscript is up-to-date. Even though there is an obvious problem with the English language and the way sentences are formed (this should be corrected by a native English speaker), the manuscript is generally well-written. References used in the manuscript are carefully selected, and the general impression is that this will be a useful review article.

Reviewer 4 Report

In this review, the  authors summarized the application of EPR in electrochemical energy storage system in recent years, and mainly focused on the characterization of transition metal oxide cathode and lithium metal andoe electrode materials. The EPR is a powerful means to explore the reaction process and mechanism of the rechargeable batteries for better development of energy storage device, and this review is suitable for publishing on magnetochemistry. The authors should refine the conclusion part and give more suggestions on EPR technology for energy storage system.

Round 2

Reviewer 1 Report

The last sentence of the abstract is still present in the manuscript, even though in the response to the reviewer it is agreed that it should be deleted.

Otherwise the other necessary corrections have been addressed, and the manuscript can be accepted once the offending sentence is removed.

There are also some minor errors which should be pick up during copy-editing. 

Reviewer 3 Report

After reading the 4 reports and the answers given by the authors, I think the manuscript can be accepted as it is. The article should be considered as a mini-review of recent results on the application of EPR to batteries.  
